# Effects of Physical Exercise Intervention on Psychological and Physical Fitness in Lymphoma Patients

**DOI:** 10.3390/medicina55070379

**Published:** 2019-07-16

**Authors:** Francesco Fischetti, Gianpiero Greco, Stefania Cataldi, Carla Minoia, Giacomo Loseto, Attilio Guarini

**Affiliations:** 1Department of Basic Medical Sciences, Neurosciences and Sense Organs, School of Medicine, University of Study of Bari, 70124 Bari, Italy; 2Haematology Unit, IRCCS Cancer Institute “Giovanni Paolo II”, 70124 Bari, Italy

**Keywords:** cancer, self-efficacy, fatigue, special population, adapted physical activity

## Abstract

*Background and objectives:* Lymphoma patients experience a psychological and physiological decline that could be reversed by exercise. However, little is known about the effects of the exercise on psychological and physical fitness variables. Therefore, the purpose of this longitudinal study was to assess self-efficacy, fatigue and physical fitness before and after an eight-week exercise intervention. *Materials and Methods:* Thirty-six participants (54.4 ± 19.1 years) performed a supervised exercise program (~60 min, 2d·wk^−1^). Each session included a combined progressive training of cardiorespiratory, resistance, flexibility and postural education exercises. Self-efficacy and fatigue were measured with the Regulatory Emotional Self-Efficacy scale and 0–10 rating scale, respectively. Physical fitness was assessed with the body mass index, lower back flexibility, static balance, muscle strength and functional mobility. *Results:* Adherence to exercise was high (91.2% ± 4.8%) and no major health problems were noted in the patients over the intervention period. At baseline, significant differences were found between Hodgkin’s lymphoma and non-Hodgkin’s lymphoma patients by age and all dependent measures (*p* < 0.05). Fatigue significantly decreased and the perceived capability to regulate negative affect and to express positive emotions improved after exercise (*p* < 0.001). Significant improvements were found for body mass index, trunk lateral flexibility, monopodalic balance, isometric handgrip force and functional mobility (*p* < 0.001). Fatigue was significantly correlated with handgrip force (*r* = −0.56, *p* < 0.001) and functional mobility (*r* = −0.69, *p* < 0.001). *Conclusions:* The supervised exercise program improved psychological and physical fitness without causing adverse effects and health problems. Therefore, exercise to improve fitness levels and reduce perceived fatigue should be considered in the management of lymphoma patients.

## 1. Introduction

Cancer is an important public health concern worldwide. Among the various types of cancer, lymphomas are very common [1]. Lymphomas are cancers of the lymphatic system and the main subdivision is made between Hodgkin’s lymphoma (HL) and non-Hodgkin’s lymphoma (NHL) [1]. Lymphoma patients are often treated multiple times with chemotherapy, radiation therapy, and/or biologic therapy interspersed with active surveillance. Hence, cancer patients and survivors bring with them physiological and psychological side effects including muscular atrophy, weight changes, lowered aerobic capacity, decreased strength and flexibility, depression, fatigue, nausea, and an overall decrease in quality of life [2,3,4,5].

Among all the side effects, cancer-related fatigue is the most common. It differs from normal fatigue of everyday living activities and affects up to 70% of cancer patients during chemotherapy and radiotherapy, as well as after surgery [4,6]. Moreover, the sedentary habits usually recommended by the biomedical staff and the family to protect the patient may lead to the development of a self-perpetuating fatigue cycle [4], which results in higher and higher levels of catabolic processes at all levels (i.e., physical, emotional, social). Exercise breaks this downward cycle and diminishes cancer-related fatigue [4]. Indeed, studies in patients with lymphoma found that changes in fatigue were largely explained by changes in physical fitness variables [2,7].

Physical activity has been demonstrated to play a preventive role in terms of the risk of developing cancer [8,9] and is considered as a major tool to improve the quality of life and survival of patients with cancer [10]. Data show that higher levels of physical activity are associated with lower overall cancer mortality [1,11]. Exercise, both during and after treatment, is an effective tool to improve functional capacity, muscular strength, functional mobility (i.e., improving balance will lower the risk of falls and fractures), fatigue, psychological well-being (i.e., reducing the risk of anxiety and depression), and health-related quality of life in cancer patients and survivors [12,13,14,15]. Some authors have even reported that exercise can improve the survival rate after diagnosis of breast cancer [16,17] and prostate cancer [18,19]. However, the benefits of physical exercise may vary according to the type of cancer and treatment; the stage of disease; the mode, intensity, and duration of the exercise program; and the current lifestyle of the patient [20]. There is still much research needed in this area.

Some studies have evaluated the benefits of strength training and combined aerobic and strength training in cancer patients and survivors, reporting enhancements in many areas, including functional mobility [21,22], flexibility [23,24], fatigue [25,26] and psychological well-being [23,27], which were also analyzed in the present study. However, in the literature few longitudinal studies could be found for psychological and physical fitness in lymphoma patients [2,28] and cross-sectional studies indicated that only 21% to 29% of lymphoma survivors [29,30] meet the public health guidelines for physical activity defined by the American College of Sports Medicine. Therefore, the aim of this study was to investigate the effects of an eight-week combined (cardiorespiratory, resistance and postural) exercise intervention program on psychological and physical fitness measures in lymphoma patients. We hypothesized that the perceived self-efficacy and fatigue, body mass, lower back flexibility, balance, muscle strength and task specific functional mobility would improve with exercise.

## 2. Materials and Methods

### 2.1. Research Design

This study used a single-group observational longitudinal research design to collect data before and after an eight-week treatment and compare the difference between pre-test and post-test data. This design was adopted because all eligible cancer patients had to be trained as soon as possible. Preliminarily, a pilot study was carried out between the months of October and November 2018. Fifteen participants (age: 56.6 ± 16.6 years; BMI: 25.2 ± 5.0 kg·m^−2^; 4 males and 11 females) affected by different forms of cancer (breast cancer, Hodgkin’s lymphoma, non-Hodgkin’s lymphoma, multiple myeloma, colon cancer, and polycythemia vera) showed significant improvements in perceived self-efficacy and cancer-related fatigue, body mass, lower back flexibility, static balance and task specific functional mobility after 8 weeks of a physical exercise intervention program. Improvements (*p* < 0.05) ranged from 12.8% to 66.7% with a moderate to large effect size. Following the positive results obtained by the pilot study, we strengthened the study design by recruiting more participants and delimited the research to lymphoma patients. Research assistants, that is, exercise professionals, with no knowledge of the research aims were trained in standardized testing procedures and designated to measure the dependent variables.

### 2.2. Participants

Before study participation, informed consent was provided by each participant. This study did not involve human individuals from a clinical or therapeutic point of view. A human sample was used, without medical contraindications, to examine only the influence of physical exercise as an educational means to improve lifestyles and self-efficacy. The study was conducted in accordance with the Declaration of Helsinki, and the protocol was approved by the Ethics Committee of IRCCS National Cancer Institute Bari, Italy (Registration number: 553/EC; date: 14/12/2015).

A preliminary screening for participant selection was carry out in the database of the Oncology Department, “Haematology Unit”. Participants were contacted by telephone (from October to November 2018) and a preliminary medical examination and the completion of a lifestyle history questionnaire were performed before participating in the study (December 2018). After the corresponding oncologist gave consent, the participants were considered suitable for the study if they met the inclusion criteria and did not present contraindications to physical exercise for patients with cancer. The inclusion criteria were as follows: (1) Histologically proven diagnosis of lymphoid neoplasm including indolent lymphoma, or Hodgkin’s lymphoma; and (2) age ≥ 18 and < 80 years. The exclusion criteria were as follows [31]: (1) Hemoglobin < 10.0 g × dL^−1^; (2) white blood cells < 3000/mL; (3) neutrophil count < 0.5 × 10^9^ x mL^−1^; (4) platelet count < 50 × 10^9^ mL^−1^; (5) fever > 38 °C; (6) unsteady gait (ataxia); (7) cachexia or loss of >35% of premorbid weight; (8) limiting dyspnea with exertion; (9) bone pain; (10) severe nausea; (11) extensive skeletal metastases.

The data from the pilot study allowed us to estimate the sample size [32], which was calculated with an assumed type I error of 0.05 and a type II error rate of 0.10 (90% statistical power); it was suggested that 19 participants would be sufficient to observe moderate effect sizes in the difference between two dependent means. However, among contacted *(n =* 59; 24 males and 35 females), 41 participants (15 males and 26 females) met all the above-mentioned eligibility criteria. Five patients refused to participate (3 males and 2 females) and finally, 36 patients (12 males and 24 females) accepted to participate and were assigned to exercise groups (mean age, 54.4 ± 19.1 years; body mass, 68.3 ± 12.1 kg; body height, 1.62 ± 0.1 m). Among the cancer diseases diagnosed are included: Non-Hodgkin’s lymphoma (NHL, n = 20; Therapy, n = 12) and Hodgkin’s lymphoma (HL, n =16; Therapy, n =12). Patients received medical treatment according to international standards and institutional guidelines. The study was carried out between the months of January and March 2019. All participants completed the study. Multiply sign shall be 5×, not 5.

### 2.3. Testing Procedures

Measurements were made at baseline (pre-test) and after 8 weeks (post-test). First, the anthropometric measurements were collected. Body height (in cm to the nearest 0.1 cm) was measured using a SECA^®^ stadiometer, and body weight (in kg to the nearest 0.1 kg) was measured using Tanita^®^ digital scales. The patients were barefooted and wore light clothing during the measurements. Body mass index (BMI) was calculated as body weight (kg) divided by the square of body height (m^2^).

Next, the following psychological and physical fitness measures were collected: (1) Perceived self-efficacy (RESE scale), (2) cancer-related fatigue (0−10 subjective rating scale), (3) lower back flexibility (trunk lateral flexibility test), (4) static balance (stork balance stand test), (5) muscle strength (isometric handgrip force), and (*6*) functional performance (30 s chair stand test). All participants were tested and trained in a gym located inside the oncology institute. One week before pre-test, two familiarizations sessions were held. Initial and final test measurements were made at the same time of day and under the same experimental conditions. All measurements were performed and supervised by the same exercise professionals, that is Adapted Physical Education Specialists.

#### 2.3.1. Regulatory Emotional Self-Efficacy (RESE) Scale

Self-efficacy beliefs influence self-regulative standards adopted by people, whether they think in an enabling or a debilitating manner, the amount of effort they invest, how much they persevere in the face of difficulties, and their vulnerability to stress and depression. On the basis of this reasoning, we administered to cancer patients an instrument to assess self-efficacy in regard to emotional regulation and, in particular, perceived self-efficacy in managing negative affect in response to adversities or frustrating events and in expressing or managing positive emotions such as joy, enthusiasm, and pride [33,34,35]. Participants rated (ranging from 1 (not well at all) to 5 (very well)) their capability to manage their emotional life with the RESE scale. This scale included items on perceived capability to regulate negative affect (8 items; α = 0.90) and to express positive affect (7 items; α = 0.78).

#### 2.3.2. Cancer-Related Fatigue Subjective Rating

Fatigue is a symptom affected by multiple biological and psychosocial factors. When assessing cancer-related fatigue, therefore, we need to include both subjective and objective data. To assess cancer-related fatigue, the participants were asked two questions [36] to help assess the severity of fatigue and its effect over time: (1) Are you experiencing any fatigue? (2) If so, how severe has it been, on average, during the past week? (If fatigue is present a simple 0–10 rating scale can be used, that is, 0–3 is mild fatigue, 4–6 moderate, and 7–10 severe). All patients became familiar with this scale before the study and followed standardized instructions to assess perceived exertion. Scores was collected and recorded before and after the eight-week intervention period.

#### 2.3.3. Trunk Lateral Flexibility Test

Trunk flexibility is important in the ability to carry out activities of daily living. Estimates of trunk flexibility frequently affect diagnostic, prognostic, and therapeutic decisions for a variety of health disorders [37]. Each participant was measured for trunk lateral range of motion using a tape measure. The same tape measurement procedure has been reported previously and has high levels of reliability with repeated measures (ICC = 0.98) [38]. Participants first underwent a 15 s static stretch in the lateral trunk motion and then were tested. Participants stood on the floor with arms in the neutral position, heels together, knees and back straight. Then they bent toward the right/left with elbow and fingers straight and attached hand on their lateral side of leg. The distance (cm) between the tip of third finger and the floor was measured three times and the lower measure was used in the analyses. The test-retest reliability reported a high reliability for this test (ICC = 0.98).

#### 2.3.4. Stork Balance Stand Test

This test evaluates postural static balance [39]. Participants were tested on the dominant and non- dominant leg. The participants were instructed to lift and hold the contralateral leg against the medial side of the knee of the stance leg while keeping his hands on the iliac crests. The trial ended when the heel of the involved leg touched the floor, the hands came off the hips, or the opposite foot was removed from the stance leg. This test was conducted with eyes opened only. The participants performed three attempts and the best time (section.) was recorded for analysis. High test-retest reliability has been reported for this test with an intraclass correlation coefficient of 0.92.

#### 2.3.5. Muscle Strength

Isometric handgrip force was measured using a digital hand dynamometer (Trailite, LiteXpress, Ahaus, Germany) [31]. Peak hand-grip force was assessed at the right side with the elbow at 90° of flexion and forearm and wrist in a neutral position. The test was performed three times and the best result was used for further analysis. The result was expressed in kilograms. The test-retest reliability reported a high reliability for this test (ICC = 0.97).

#### 2.3.6. Thirty Seconds Chair Stand Test

This test is one of the most important functional evaluation clinical tests because it measures lower body strength and relates it to the most demanding daily life activities (e.g., climbing stairs, getting out of a chair or bathtub, or rising from a horizontal position) [40]. It is also able to assess functional fitness levels [41] and the fatigue effect caused by the number of sit-to-stand repetitions. It consists of standing up and sitting down from a chair as many times as possible (*n*) within 30 s. A standard chair (with a seat height of 40 cm) without a backrest but with armrests was used. Initially, the participants were seated on the chair with their back in an upright position. They were instructed to look straight forward and to rise after the “1, 2, 3, go” command at their own preferred speed with their arms folded across their chest. All trials were performed using the same chair and with similar ambient conditions.

### 2.4. Exercise Intervention

Currently, there is no evidence supporting a different training response to exercise in a patient with cancer from that in the general adult population. Accordingly, in the present study, the American College of Sports Medicine guidelines for cancer survivors were followed [42]. All sessions were conducted in small groups of five participants under direct supervision of exercise professionals that are specialists in Adapted Physical Education to ensure safety, proper intensity, and appropriate exercise technique. Additionally, the mode, frequency, intensity, duration, and progression in an individual exercise log were recorded to ensure adequate training. For the intervention period all participants were asked not to perform any physical activity outside the oncology institute gym.

After data collection, the intervention group performed an eight-week program consisting of twice-per-week exercise sessions, each lasting 60 min on average. Every single exercise session was divided into a 10-min warm-up (i.e., postural education exercises and stretching of all major muscle groups), a 40-min main exercise period (i.e., aerobic exercise, resistance training), and a 10-min cooldown period (i.e., stretching again and/or postural education exercises).

During the main exercise period, cardiorespiratory training consisted of progressive 30-min of walking and stationary bike at an intensity that ranged from 45% to 75% of heart rate reserve. Heart rate (HR) was monitored by the participants and the exercise professionals during training using a Polar HR monitor (Target model, Kempele, Finland). Aerobic exercise duration was initially 20 min and was divided into equal parts between the two exercise modes following a rotational order. Based on the recommendations in the literature [10,42], the aerobic-exercise period was increased by 2 min a week, such that it was 34 min during week 8. However, ≥150 min/week of moderate intensity or ≥75 min/week of vigorous intensity were recommended by ACSM [42]. Unfortunately, in this study it was not possible to reach this goal for several reasons (i.e., short duration of the study and low initial fitness level of the patients).

Resistance training consisted of progressive 10–20 min exercises with free weights and/or resistance bands, at an intensity ranging from 50% to 70% of 1RM [37] for lifts involving the lower body and from 40% to 70% of 1RM for lifts involving the upper body. A total of 8–10 exercises for major muscle groups (i.e., chest, back, shoulders, triceps, biceps, abdominals, lower limbs muscles), 1 or 2 sets of 8 to 12 reps, and a rest 1–3 min between exercises and sets, were performed with a gradual increase in resistance (1–2 kg) following two consecutive symptom-free sessions [10].

Flexibility was trained before and after the main period by stretching exercises performed maximally on all major muscle groups (1–3 sets per muscle group) but avoiding pain, especially in joints. Duration was gradually increased from 10 to 30 s per stretch, repeating one to three times for a total of 60 s per stretch. Following approval from a surgeon, special attention was given to shoulder mobility stretches in breast cancer survivors.

Postural education exercises were carried out both in the warm-up and cooldown period, and consisted of breathing, proprioception and balance exercises. In some sessions a pilates mat workout has also been integrated.

Finally, the exercise program focused on physical activities that use large muscle groups rather than small groups, since most daily living tasks depend on these large muscle groups. Session design and exercises were modified according to the acute or chronic treatment effects of surgery, chemotherapy, or radiotherapy.

### 2.5. Statistical Analysis

Normality of all parametric variables was tested using Shapiro-Wilk test procedure. A paired sample t-test was used to determine if the changes from pre- to post-test for the parametric variables were statistically significant. For non-normally distributed data, the Wilcoxon signed-rank test was used. The independent t-test was used to examine differences in baseline parameters between HL and NHL. For non-normally distributed data, the Mann-Whitney U test was used. To examine correlations between the different parameters, the Spearman rank test was used.

For parametric data, the effect size (ES) was calculated as the post-training mean minus pre-training mean divided by pooled SD before and after training and was interpreted as small (0.20–0.49), moderate (0.50–0.79) and large (≥ 0.80) effect. For nonparametric data, ES was determined by dividing the z value by the square root of N and interpreted as small (0.10–0.29), moderate (0.30–0.49) and large (≥ 0.50) effects [43].

The reliabilities of the physiological measures were assessed using the intraclass correlation coefficients; scores from 0.8 to 0.9 were considered as good, while values >0.9 were considered high [44]. To assess the internal consistency of the psychological measures, Cronbach’s alpha was used; scores from 0.70 to 0.79 were considered reliable, from 0.80 to 0.90 as highly reliable, and >0.90 as very highly reliable [45]. Percentage changes were calculated as ((post-training value – pre-training value)/pre-training value) × 100.

All analyses were conducted with SAS JMP^®^ Statistics (Version < 14.3 >, SAS Institute Inc., Cary, NC, USA, 2018). Parametric data are presented as group mean values and standard deviations, and categorical data as median and minimum and maximum. An alpha level of *p* < 0.05 was considered statistically significant.

## 3. Results

Patients characteristics and baseline measurements are reported in Table 1. Thirty-six patients were included in study and twenty were diagnosed with NHL (55.6%), sixteen with HL (44.4%). Mean age was significantly lower in HL patients than in NHL patients (*p* < 0.001).

Adherence to the supervised exercise program averaged 91.2% ± 4.8%. No adverse effects and no health problems were noted in the participants over the eight-week period. Participants were satisfied with the results of the study and reported wanting to continue the training program on their own. Changes and statistical data in the psychological and physical fitness values over the eight-week intervention program are reported in Table 2.

### 3.1. Psychological Measures

At baseline, HL patients reported a greater perceived self-efficacy in managing negative affect in response to adversities (*p* < 0.001) and in expressing positive emotions (*p* = 0.021) compared to NHL patients. In addition, the perceived cancer-related fatigue was significantly lower in the HL patients than NHL patients (*p* = 0.001) (Table 1).

Statistical analysis revealed that measures of fatigue significantly decreased (*p* < 0.001) in the exercise group between the pre and post-study measurements. Instead, their capability to manage their emotional life has improved both in the perceived capability to regulate negative affect (*p* < 0.001) and to express positive emotions (*p* < 0.001) (Table 2).

### 3.2. Physical Fitness Measures

At baseline, the body mass index was significantly lower in HL patients compared to NHL patients (*p* = 0.009). In addition, the HL patients showed significantly higher values than NHL patients in the trunk lateral flexibility (Left: *p* = 0.015; Right: *p* = 0.002), stork balance stand (Left: *p* = 0.011; Right: *p* < 0.001), isometric handgrip force (*p* < 0.001) and 30 s Chair Stand Tests (*p* = 0.036) (Table 1).

Over the eight weeks of treatment, the BMI significantly decreased (*p* < 0.001). In addition, highly significant increases in physical and functional fitness measures were observed. Paired sample t-tests detected a significant decrease in the distance between the tip of third finger and the floor in the trunk lateral flexibility test (Left: *p* < 0.001; Right: *p* < 0.001), and significant increases of the isometric handgrip force (*p* < 0.001) and the number of standing up and sitting down from a chair within 30 s (*p* < 0.001). Finally, significant gains in seconds in the stork balance stand test (Left: *p* < 0.001; Right: *p* < 0.001) were found by the Wilcoxon signed-rank test (Table 2).

### 3.3. Correlations between Psychological and Physical Fitness Parameters

After the eight-week intervention program, relevant and significant correlations were detected between handgrip force and fatigue (*r* = −0.56, *p* < 0.001), handgrip force and 30 s chair stand test (*r* = 0.34, *p* = 0.04), fatigue and 30 s chair stand test (*r* = −0.69, *p* < 0.001), fatigue and right leg balance (*r* = −0.50, *p* = 0.002), 30 s chair stand test and perceived self-efficacy in expressing positive emotions (*r* = 0.38, *p* = 0.024).

## 4. Discussion

Considering the health benefits derived from physical exercise in cancer patients and the lack of longitudinal studies that investigate the effects of physical fitness in lymphoma patients, this study examined the effects of combined aerobic, resistance and postural exercises on the psychological and physical fitness variables. It was found that after eight weeks of exercise intervention, the perceived self-efficacy and cancer-related fatigue, body mass, back flexibility, static balance, muscle strength and specific functional mobility significantly improved in patients with lymphoma. Furthermore, we were able to observe the age difference between patients with NHL and HL at baseline, which consequently influenced the psychological and physical fitness performance in the tests. This finding confirms the age-related incidence, that is, HL has a first high peak in the third decade of life and a much smaller peak occurring after the age of 50, whereas in NHL the incidence increases with increasing age [1].

This study’s results are in agreement with several other studies that showed the benefits of combined aerobic and strength training in cancer patients and survivors, reporting improvements in functional fitness levels and psychological well-being [21,22,23,24,25,26,27]. Many studies demonstrated that exercise performed both during and after treatment is an effective tool to achieve health benefits in terms of functional performance, fatigue, psychological wellbeing, and health-related quality of life in cancer patients and survivors [7,12,13,14,15,46,47]. However, it is known that the benefits of physical exercise may vary according to the type of cancer and treatment, and the current lifestyle of the patient [20]. For this reason, one of the strengths of our longitudinal study is the decrease in fatigue in patients with lymphoma, explained by improvements in fitness variables [2,7]. This was also confirmed by the significant negative association between fatigue and handgrip force, and between fatigue and the 30 s chair stand test. Another strength of this study is the increase in ability, on the part of lymphoma patients, to manage their emotional life. In particular, they improved the emotional perceived self-efficacy in managing negative affect in response to adversities or frustrating events and in expressing positive emotions such as joy, enthusiasm, and pride [33,34,35]. The significant positive association between the 30 s chair stand test and RESE scale could indicate a greater perceived self-efficacy in managing positive emotion by the patients that have high functional fitness levels. These are notable results because cancer patients and survivors bring with them physiological and psychological side effects, including inter alia, vulnerability to stress and depression [3,23].

Improvements with moderate/large effect sizes demonstrated the effectiveness of the exercise intervention program on body mass index, lower back flexibility, static balance, muscle strength and functional mobility. After intervention, the body mass index reduction was a positive result, as it is known that weight gain is a common side effect of cancer therapy along with other physiological and psychological side effects [2,3,4,5]. These findings agree with exercise interventions that have reported improvements in body mass in cancer patients and survivors [48,49]. Weight management is crucial for these people because weight gain is associated with higher cancer mortality rates and avoiding weight gain could offer a greater chance of surviving cancer [17]. In this short intervention study, although improvements in body mass were minimal, the results gave us hope for future long-term studies. In addition, improvements in flexibility, balance and muscle strength may help the patients to perform the activities of daily life more easily without being overwhelmed by fatigue. In fact, our combined exercise program not only allowed to increase the functional and physical fitness levels of all lymphoma patients, as well as the fatigue resistance, but also to reduce the subjective perception of fatigue. Thus, the combined results of the increase of the lower body strength and endurance shown in the 30 s chair stand test could explain the reduced levels of perceived fatigue observed in the lymphoma patients [2,7]. In this regard, we have given lymphoma patients the opportunity to improve their quality of life, as demonstrated by previous studies [12,13,14,15], and to understand the importance of the physical activity as a major prevention tool [29,30].

No adverse effects and no health problems were found in the participants during the eight-week period, demonstrating the effectiveness of our combined exercise intervention program. Participants were satisfied with the results of the study and reported their intention to continue the training program on their own. Therefore, this study demonstrated that a few weeks of regular exercise could be sufficient to help patients deal with anti-cancer treatment and its deleterious side effects [46,50,51]. However, these results could be the consequence of the high level of deconditioning of cancer patients, such that any small stimulus, such as a short exercise program, may lead to the partial recovery of the patient’s normal physiological and psychological characteristics. More work is needed to elucidate the long-term beneficial effects of combined exercise training in cancer patients and, specifically, in lymphoma patients.

Our study presents several weaknesses must be known. First, the relatively small number of patients divided by gender and the absence of a control group indicate that the study may not allow valid conclusions to be drawn. However, the participants number was calculated on the basis of the previous pilot study, which showed improvements in the dependent measures with a moderate to large effect sizes. In the present study, this number was greatly exceeded. We experienced difficulties with the inclusion of patients, which is inherent in studying this patient population. The explanations of these difficult inclusions were similar to those reported in other studies on cancer patients [2,52]. These explanations include a difficult programming of the measurements before the start of the intervention, the patient’s emotional state following diagnosis and the debilitating effect of the disease on the patient; this makes testing and training complex. The reasons for initial dropout were similar to those reported in other studies in cancer patients [2,52]. A second limitation is the predominance of female patients in our study sample. This gender imbalance is mainly a consequence of the predominance of females in eligible patients during the inclusion period. It cannot be discounted that this gender imbalance has led to bias in the results. Finally, the short duration of the study does not allow us to know the effects of the exercise in the long term. However, short-term studies could have an advantage, since cancer sufferers must enter into training programs immediately.

## 5. Conclusions

This short-term intervention study aimed to investigate the effect of combined aerobic, resistance and postural exercises on both the psychological and physical fitness in lymphoma patients. Results suggested that exercise could improve the perceived cancer-related fatigue, confirming the physical exercise as a major prevention tool, but also as capable of improving the emotional perceived self-efficacy in managing negative affect and in expressing positive emotions. Furthermore, body mass, physical fitness and functional capacity were enhanced, providing an important support to lymphoma patients undergoing treatment. In this way, it may actually be possible to prevent and minimize physical inactivity, fatigue, muscle wasting and loss of energy. However, more research is needed on the optimal type, intensity, duration, and frequency of exercise for the health and wellbeing benefit of this special population.

## Figures and Tables

**Table 1 medicina-55-00379-t001:** Baseline characteristics and measurements of study participants.

	All Patients	NHL	HL
Number of patients	36 (100%)	20 (55.6%)	16 (44.4%)
Men, *n*	12 (33.3%)	4 (20%)	8 (50%)
Age (years)	54.4 (19.1)	68.2 (7.9)	37.3 (14.2) ^‡^
**Psychological measures**			
RESE scale Negative (8–40)	25 (16–35)	21 (16–26)	30.5 (25–35) ^‡^
RESE scale Positive (7–35)	28 (22–35)	27 (22–30)	31 (26–35) ^*^
Fatigue rating scale (0–10)	5 (4–5)	5 (4–5)	4 (4–5) ^†^
**Physical fitness measures**			
Body mass index (kg·m^−2^)	25.9 (4.9)	27.8 (5.2)	23.7 (3.5) ^†^
Trunk flexibility L (cm)	13.4 (3.0)	12.4 (3.7)	14.6 (0.4) *
Trunk flexibility R (cm)	15.3 (3.6)	13.8 (3.8)	17.1 (2.2) ^†^
Stork balance L (s)	4.3 (1.1–49)	2 (1.3–15)	39 (1.1–49) *
Stork balance R (s)	2.3 (1.9–43)	2.2 (1.9–13.8)	30 (2.3–43) ^‡^
Handgrip force (kg)	39.2 (6.0)	34.5 (3.1)	45.1 (2.1) ^‡^
30 s chair (n)	14.7 (3.2)	13.6 (2.3)	16.0 (3.8) *

Notes: Normally distributed parameters expressed as mean (SD) and non-normally distributed parameters as median (range). Abbreviations: NHL, non-Hodgkin’s lymphoma; HL, Hodgkin’s lymphoma; RESE, Regulatory Emotional Self-Efficacy; L, left; R, right. * *p* < 0.05, † *p* < 0.01, ‡ *p* < 0.001 HL versus NHL.

**Table 2 medicina-55-00379-t002:** Effects of an eight-week supervised exercise program in lymphoma patients.

Variables	Baseline	Post-Exercise	ES (Interpretation)	Difference	
**Psychological**				Absolute	%
RESE scale Negative (8–40)	25 (16–35)	31 (27–35) ^‡^	1.11 (large)	6.0 (3.9)	+24.0
RESE scale Positive (7–35)	28 (22–35)	33 (29–35) ^‡^	1.36 (large)	4.0 (2.7)	+14.0
Fatigue rating scale (0–10)	5 (4–5)	4 (2–4) ^‡^	1.54 (large)	−1.2 (0.4)	−26.3
**Physical fitness**					
Body mass index (kg·m^−2^)	26.0 (4.9)	25.1 (4.0) ^‡^	0.58 (moderate)	−0.8 (1.0)	−3.1
Trunk flexibility L (cm)	13.4 (3.0)	12.0 (2.6) ^‡^	0.89 (large)	−1.4 (1.2)	−10.5
Trunk flexibility R (cm)	15.3 (3.6)	13.7 (2.9) ^‡^	0.83 (large)	−1.6 (1.4)	−10.5
Stork balance L (s)	4.3 (1.1–49)	4.6 (1.8–63) ^‡^	0.80 (large)	7.0 (8.7)	+43.9
Stork balance R (s)	2.3 (1.9–43)	7.9 (3–58) ^‡^	1.07 (large)	9.7 (9.1)	+73.8
Handgrip force (kg)	39.2 (6.0)	41.4 (6.1) ^‡^	1.12 (large)	2.3 (1.3)	+5.9
30 s chair (n)	14.7 (3.2)	17.6 (3.7) ^‡^	1.56 (large)	2.9 (0.9)	+19.8

Notes: Normally distributed parameters expressed as mean (SD) and non-normally distributed parameters as median (range). Absolute and percentages differences were calculated using the group’s mean values. Abbreviations: RESE, Regulatory Emotional Self-Efficacy; L, left; R, right. ES, effect size; ^‡^
*p* < 0.001 baseline versus post-exercise.

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
