# Peer review of "Effects of Physical Exercise Intervention on Psychological and Physical Fitness in Lymphoma Patients"

_medicina, 2019, doi:10.3390/medicina55070379_

Round 1

Reviewer 1 Report

Thank you for your revisions based on the previous review. This is much improved. 

Comments to Editors are as follows:

The fundamental issues with this paper (i.e. lack of control group, low patient number) cannot be easily addressed. In this revised form it is an interesting paper which does have something to add to the literature in this field.

Currently the result section is the major weakness as it is both brief and repetitive. The table and figure show the same data set so both are probably not needed.

Minor type/spelling errors are present in most sections.

Author Response

Responses to the reviewer 1

Reviewer: The fundamental issues with this paper (i.e. lack of control group, low patient number) cannot be easily addressed. In this revised form it is an interesting paper which does have something to add to the literature in this field.

Authors: The lack of the control group is justified by the fact that all patients need to be trained (no wait list). This was highlighted in the "Research design" section (Lines 80-81). The number of the patients required was calculated by an a priori statistical power analysis (Lines 111-118; 19 participants required). In our study we used a larger number of participants needed for the study (36 vs. 19 patients). Also, this was highlighted in “Discussion” section (lines 365-369).

Reviewer: Currently the result section is the major weakness as it is both brief and repetitive. The table and figure show the same data set so both are probably not needed.

Authors: We are aware that we must not repeat the same data in the graphs, tables and text in the "Results" section. However, histograms can make the study more understandable to a greater number of readers. Dissemination work is essential for a researcher! In any case, we have removed the figure 1 because it shows the same data of Table 2.

We  consider the results complete, as all the necessary statistical calculations are included, including the effect size and the correlations, as well as all the descriptive data.

Reviewer 2 Report

The study is interesting. That is why it should be done with greater methodological rigour.

The introduction is very generic, focusing little on the object of study which is the effect of exercise on some psychological aspects and functional capacity.

The methodology of the intervention is poorly explained. For example: how was 1RM evaluated in order to be able to prescribe the training load with elastics or free weight? which muscle groups were exercised? which were the exercises performed for strength training? How was the progression?

The duration and intensity of the load does not reach the minimum recommended by WHO.

There is no need to make a comparison (let alone draw conclusions) between the HL group and the NHL group, as they are of different ages and with different male/female ratios. It makes no sense to study the correlation between physical evidence, nor through the ICC, why and for what is it done and more in the context of this study?

The absence of reference bibliography for psychological tests is striking, and in the discussion they are not confronted with other studies that have used these tests.

The conclusions are not plausible, little founded, little discussed. Do you think that the statistically significant changes were also clinically relevant; are the tests you have used specific and sensitive to the component of physical condition that you theoretically evaluate? With these three tests, can an extrapolation of physical condition be made? It cannot be attributed from a plausible perspective that exercise has helped them to lower BMI (they did not control confounding factors, intake and energy expenditure), and furthermore, as indicated in the introduction, these patients tend to lose weight.

Author Response

Responses to the reviewer 2

Reviewer: The introduction is very generic, focusing little on the object of study which is the effect of exercise on some psychological aspects and functional capacity.

Authors: We do not agree on this point because the introduction contains a review of very important studies from high-impact journals, they were written in a clear and concise way, describing what is known and what is not known and the hypothesis is formulated on the basis of 30 bibliographical references !! This shows that the introduction is not absolutely generic but focused on the psychological and physiological variables we have studied.

Reviewer: The methodology of the intervention is poorly explained. For example: how was 1RM evaluated in order to be able to prescribe the training load with elastics or free weight? which muscle groups were exercised? which were the exercises performed for strength training? How was the progression?

Authors: For the evaluation of the 1RM we added a reference (36). The major muscle groups are in the public domain; however, we have indicated in the text the muscle groups involved for the strength training (lines 223-224) and added a reference (10). The progression was already indicated in text (lines 225-226) but we also added a reference (10).

Reviewer: The duration and intensity of the load does not reach the minimum recommended by WHO.

Authors: Yes. For more reasons, that is for the short duration of the study and the low initial fitness level of the patients, it was not possible to reach the goal recommended by the international guidelines (added text from line 217 to 220).

Reviewer: There is no need to make a comparison (let alone draw conclusions) between the HL group and the NHL group, as they are of different ages and with different male/female ratios. It makes no sense to study the correlation between physical evidence, nor through the ICC, why and for what is it done and more in the context of this study?

Authors: We don't agree! It is very important to compare HL and NHL patients, at least at baseline, as they are people with different clinical conditions. The age difference found confirms the age-related incidence (lines 314-318). The gender imbalance is instead commented in the "discussion" section (lines 375-377).

The ICC means intraclass correlation coefficient and is a fundamental statistical calculation to verify the reliability of the tests, as well as the Cronbach's alpha is used to evaluate the internal consistency of the items. We don't understand why they don't make sense! Math as well as statistics are not an opinion!

The correlation between handgrip force and fatigue and the other correlations we consider them important because they provide us with further data to support our experimental hypothesis.

Reviewer: The absence of reference bibliography for psychological tests is striking, and in the discussion,  they are not confronted with other studies that have used these tests.

Authors: We do not agree with this comment! References 33 to 35 represent the study that validated the psychological tests used. These tests are still current and are used in many other studies to assess perceived self-efficacy and fatigue. We are the first to have used the Caprara test (33,34) to assess self-efficacy on cancer patients, as we consider it a valid and effective test and above all practical to be administered to this type of population. In addition, psychological tests used in this study have a high concurrent validity. Self-efficacy and fatigue were well-discussed in “Discussion” section.

Reviewer: The conclusions are not plausible, little founded, little discussed. Do you think that the statistically significant changes were also clinically relevant; are the tests you have used specific and sensitive to the component of physical condition that you theoretically evaluate? With these three tests, can an extrapolation of physical condition be made? It cannot be attributed from a plausible perspective that exercise has helped them to lower BMI (they did not control confounding factors, intake and energy expenditure), and furthermore, as indicated in the introduction, these patients tend to lose weight.

Authors: In “Participants” subsection we wrote: “This study did not involve human individuals from a clinical or therapeutic point of view. A human sample was used, without medical contraindications, to examine only the influence of physical exercise as an educational means to improve lifestyles and self-efficacy”. Special issue of Medicina Journal is called: “Physical Activity and Physical Fitness in the Prevention and Control of Non-Communicable Diseases”. The exercise professionals aim to the physical and mental wellbeing in populations, and, in our study, we have improve the psychological and physical fitness in patients. The results obtained showed a high practical relevance as the effect size has been moderate to large.

The tests we used were specific and sensitive to measure physical and psychological fitness as they are validated standardized tests and have shown high reliability in this study.

In “introduction” is wrote “weight changes” (line 41) and not “lose weight”.  Instead, from line 339 to 344 we can understand that this comment lacks presuppositions. Specifically, the line 340 says: “weight gain is a common side effect of cancer therapy along with other physiological and psychological side effects [2-5]”.

For these reasons and for the reasons explained above, we consider the conclusions founded and well-discussed.

Authors’ conclusions

This is the first study that has investigated the effect of combined aerobic, resistance and postural exercises on both psychological and physical fitness in lymphoma patients. Findings demonstrated the effectiveness of our combined exercise intervention program. In addition, participants were satisfied with the results of the study and reported their intention to continue the training program on their own. The results obtained, therefore, go beyond improving the physical and mental condition, as we probably provided the input to change patients lifestyle.

Round 2

Reviewer 2 Report

We'll never agree. There is no change that would make me change my first evaluation. I do not agree with either the content of the answers or the form. I'm not getting into epistolary arguments.

Author Response

The  authors  really  appreciate  the  editor’s careful  reading  and  constructive suggestions of our manuscript (Medicina-527582). According to your recommendations, the paper was carefully revised. Major modifications are highlighted in yellow. Following are the point by point responses to the editor’s comments.

Comment 1: Methods, 2.1. Research design: Please, remove the sentence: “… and therefore no control group was needed”. I suggest the authors to confirm if a control group with no intervention is not ethically recommended. Is it this case? Please, check this interrogation.

Authors: The sentence has been removed.
After local ethics board approval and a statement from the authors declaring adherence to the Declaration of Helsinki, we asked ourselves: it is ethical to include a non-active intervention group (control) for a disease which is known to have negative effects on quality of life and survival of patients? Considered that physical activity has been demonstrated to be a major tool to improve the quality of life and survival of patients with cancer, we have therefore decided to immediately include all patients, who have met all the eligibility criteria, in the intervention program. Therefore, we confirm that a control group with no intervention is not ethically recommended for the purpose of this study. In effect, if it is known that a treatment can be effective, denying it is unethical.

Comment 2: An important point that make us confused is the time of exercise intervention. Is it possible to complete all the activities described in the methodology, (the minutes of aerobic exercise, muscle groups, series, repetitions and rest time, stretching…) in 60 minutes? Please, clarify if all steps were concluded in 60 minutes or more. 

Authors: We have made some corrections to make the exercise protocol more understandable. 
At line 206, we added “each lasted 60 minutes on average”. 
At week 1, the intervention included 10 minutes of warm-up, 30-min main exercise period (i.e., 20-min of cardiorespiratory and 10-min of resistance training) and a 10-min cooldown.  At week 1, the overall duration of the intervention was 50 minutes. 
At week 8, cardiorespiratory training lasted 34 minutes (already specified in lines 215-216) , respecting the ACSM guidelines that recommend an increase of 2 minutes per week. The duration of resistance training has been progressively increased from 10 up to 20 minutes over 8 weeks (added to line 220). The duration of the warm-up and cool-down remained unchanged over the 8 weeks. At week 8, the overall duration of the intervention was 74 minutes.

Comment 3: We would like to know whether the described procedures are possible to make as an individualized prescription of the load.

Authors: We have perfectly followed the general guidelines of the ACSM to conduct the exercise sessions. However, from the experience of the previous pilot study, here we were able to more accurately adapt the loads for each individual lymphoma patient. Once the loads to be used at the beginning of the intervention were identified, the patients subsequently showed a high response to the exercise, with a clear increase in performance, probably due to the high level of initial deconditioning. In addition, at lines 235-237 was also specified: “Session design and exercises were modified according to the acute or chronic treatment effects of surgery, chemo-therapy, or radiotherapy”. Therefore, we consider very important to follow the guidelines of the ACSM for the exercise prescription, but we needed to adapt the training loads and their progression to individual patients. Especially because little is known about the effects of exercise on lymphoma patients. However, we have tried to provide useful data and information to better understand the physiological and psychological effects of exercise on lymphoma patients.

Comment 4: Please, add important bibliographical references for some of the psychological evaluation procedures used.

Authors: The references 33,34 (Regulatory Emotional Self-Efficacy Scale) and 35 (Cancer-Related Fatigue Subjective Rating) are important references already present for the psychological assessment. However, we have added a more current reference (35) that demonstrate the multidimensionality, cross-cultural validity and cross-gender invariance of the Regulatory Emotional Self-Efficacy Scale. For this addition we have modified the numbering of references in the text from 35 on.

This manuscript is a resubmission of an earlier submission. The following is a list of the peer review reports and author responses from that submission.

Round 1

Reviewer 1 Report

Abstract:

Line 17-18 the age of participants should be after the participant number. In its current form it reads as if the exercise intervention was for 22-75 years!

No control group

Line 22 problems rather than problem

Stats analysis sentence not really needed in an abstract

P reported to a large and inconsistent number of decimal places

Introduction:

General background to cancer related fatigue is given to justify the study.

Rather too many (i.e.)’s which negatively affects the flow of the section.

The justification for short over longer duration intervention is weak.

Specific study choices in terms of exercise intervention choice and variables to be considered are not discussed or justified which makes it seem like these are not well thought through.

Materials and Methods:

No control group is used and no attempt to justify this decision is given. It could be explained on an ethical basis if needed.

Refers to subjects! Patients or participants would be more appropriate

Line 104: start not star

Reference for contraindications? Seems to suggest these are established.

Large variation in participants (age, BMI, type of cancer, therapy, diet…) with small number (n=15) and no control group means this is not a study from which we can draw a huge amount of confidence in findings. It reads like a small-scale pilot/feasibility study.

Why trunk lateral flexibility as a variable? Others are justified this is not

It is not evident that the participants physical activity outside of the training intervention was monitored in any way. This makes it difficult to attribute changes to the intervention alone and to quantify activity as a whole against the ACSM guidelines

Line 224: Effect sizes do not provide a qualitative interpretation

Results:

In some places range is reported whist SD is given in others. No suggestion of why

P reported to unusually large number of decimal places

The results section is very brief. No detail around patient’s responses to training and/or training load is included. HR/rate of perceived effort during exercise sessions  would be useful to help quantify the training done. Change in muscle mass? Mass? BMI?

Discussion:

The key findings of the study are described, and implications of the research are suggested.

Obvious limitations of the study are covered briefly.

Conclusion is a little strong given the weak study design e.g. Prevent fatigue in cancer patients.

Author Response

Responses to the reviewer 1  Comments

1. Reviewer: Line 17-18 the age of participants should be after the participant number. In its current form it reads as if the exercise intervention was for 22-75 years!

Authors: Revised and corrected.

2. Reviewer: No control group.

Authors: In this study, a single group design was used because all eligible cancer patients need to be trained and therefore a control group was not required (sentence added, lines 93-94). In effect, for research conducted in real-world settings, as in the field of physical exercise, single group design is very adopted because it could be more appropriate [1,2].

3. Reviewer: Line 22 problems rather than problem.

Authors: Revised and corrected (line 21 after correction).

4. Reviewer: Stats analysis sentence not really needed in an abstract.

Authors: Yes, we agree that it is not really necessary. We have deleted the sentence.

5. Reviewer: P reported to a large and inconsistent number of decimal places.

Authors: we presented all decimal because a low p guarantees that the null hypothesis is false. A p <0.0001 is not the same as a p <0.05. However, in this study an alpha level of p < 0.05 was considered statistically significant. Therefore, in abstract we have replaced all significant values with p<0.05, p<0.01, p<0.001, where necessary.

6. Reviewer: General background to cancer related fatigue is given to justify the study.

Authors: well.

7. Reviewer: Rather too many (i.e.)’s which negatively affects the flow of the section.

Authors: Yes, we agree. However, we believe that the studies chosen to describe and reason on this topic are necessary, as they provide a solid basis for the formulation of the hypothesis to be tested.

8. Reviewer: The justification for short over longer duration intervention is weak.

Authors: In literature, all this is well highlighted. In the manuscript are descripted studies that investigated and demonstrated the exercise effects on the physical capacity of cancer patients/survivors after short (≤ 10 weeks) training programs (line 79).Furthermore, it is well known the high level of deconditioning of cancer patients and survivors, such that any small stimulus such as a short exercise program (e.g., eight weeks) may lead to the partial recovery of  the patient’s  normal  physiological  and  psychological characteristics [3]. In addition, patients that present a severely impaired fitness may initially benefit from interrupted programs that incorporate several bouts of shorter-duration exercise [4].

9. Reviewer: Specific study choices in terms of exercise intervention choice and variables to be considered are not discussed or justified which makes it seem like these are not well thought through.

Authors: In “Introduction” were chosen studies that have assessed the benefits of resistance training and combined aerobic and  resistance training in cancer patients and survivors, reporting improvements in many areas including functional mobility, flexibility, fatigue and psychological well-being. In our study all these areas were considered as dependent variables. In addition, we were the first to attempt to identify the effect of combined aerobic and resistance training with postural education exercises.

10. Reviewer: No control group is used and no attempt to justify this decision is given. It could be explained on an ethical basis if needed.

Authors: Already answered point 2.

11. Reviewer: Refers to subjects! Patients or participants would be more appropriate.

Authors: Revised and corrected.

12. Reviewer: Line 104: start not star.

Authors: Revised and corrected.

13 Reviewer: Reference for contraindications? Seems to suggest these are established.

Authors: Reference added  in line 106.

14. Reviewer: Large variation in participants (age, BMI, type of cancer, therapy, diet…) with small number (n=15) and no control group means this is not a study from which we can draw a huge amount of confidence in findings. It reads like a small-scale pilot/feasibility study.

Authors: The problem of the absence of the control group was discussed in point 2. Voluntary participation of cancer patients in these intervention studies is very difficult, and it is even more difficult to motivate them to exercise regularly. For obvious reasons it is also difficult to have a homogeneity of the group regarding the type of cancer, therapy, diet ...However, an a priori power analysis was calculated by G*Power  [5], a software used to estimate sample size [6] and revealed that a total sample size of 12 participants would be sufficient to observe  medium effects in difference between two dependent means. Sentence added in lines 110-113.

15. Reviewer: Why trunk lateral flexibility as a variable? Others are justified this is not.

Authors: We added sentence and reference  in lines 158-160.

16. Reviewer: It is not evident that the participants physical activity outside of the training intervention was monitored in any way. This makes it difficult to attribute changes to the intervention alone and to quantify activity as a whole against the ACSM guidelines.

Authors: For this reason, all participants were asked not to perform any physical activity outside the oncology institute, for the duration of the intervention  (Sentence added in lines 194-196).

17. Reviewer: Line 224: Effect sizes do not provide a qualitative interpretation (currently line 237)

Authors: small, moderate and large are qualitative (nonmathematical) descriptors of strength of association and effect size in data interpretation, although the effect size is a standardization of the difference (difference in means in standard deviation units) and therefore it is a quantity. The term "qualitative" refers to the interpretation according to Cohen [7]. Qualitative description facilitates data interpretation and communication, particularly for the non-statistical audience: students, practitioners, novice researchers. However, we could delete the sentence if it is not deemed appropriate.

18. Reviewer: In some places range is reported whist SD is given in others. No suggestion of why.

Authors: The explanation is given in the caption of Table 1. Parametric data are shown as mean (SD), whereas categorical data (RESE and fatigue scale) are shown as median (minimum-maximum).

19. Reviewer: P reported to unusually large number of decimal places.

Authors: Already answered point 5. In addition, in Table 1 and in the "Results", we have indicated the p-value with all decimals as suggested by experts in medical statistics [8]. Therefore, we have replaced all significant values with p<0.05, p<0.01, p<0.001, where necessary.

20. Reviewer:

a) The results section is very brief. No detail around patient’s responses to training and/or training load is included. HR/rate of perceived effort during exercise sessions  would be useful to help quantify the training done.

b)Change in muscle mass? Mass? BMI?

Authors:  

a) Heart rate was monitored by the participants and the exercise professionals during training to manage the cardiorespiratory training load. Intensity ranged from 40% to 85% of heart rate reserve. The aerobic-exercise period was increased by 2 min a week, such that it was 30 min during week 8. Details are in text (lines 203-212). However, this parameter was not used as test but only to manage the training load.

The perceived effort was assessed during the test sessions through the cancer-related fatigue subjective rating (lines 147-156) and 30-second Chair Stand Test (lines 176-186). This latter test is able to assess functional fitness levels and the fatigue effect caused by the number of sit-to-stand repetitions.

b) In our study we had not planned to evaluate the effects of the training protocol on body mass, but weight measurements before and after the intervention were recorded. However, we agree that adding this new parameter could improve the quality of our research work. In the "Introduction", "methods", "results", "discussion" and “conclusions” sections, we have added the sentences concerning the evaluation of the BMI. We have also added a new figure.

21. Reviewer: The key findings of the study are described, and implications of the research are suggested.

Authors: Well.

22. Reviewer: Obvious limitations of the study are covered briefly.

Authors: We added another limitation.

23. Reviewer: Conclusion is a little strong given the weak study design e.g. Prevent fatigue in cancer patients.

Authors: The sentence has been modified stating that the exercise "could" improve the perception of fatigue (line 370).

References

1. Peacock, J.L.; Kerry, S.M.; Balise, R.R. Presenting medical statistics from proposal to publication. Oxford University Press, 2017; pp. 64-71.

2. Armstrong, L.E.; Kraemer, W.J. (2016). ACSM’s Research methods. Wolters Kluwer: Philadelphia, 2016; pp.121-142.

3. Jacobs, P. NSCA's Essentials of Training Special Populations; Human Kinetics: Champaign, IL, 2017; p. 351.

4. Ehrman, J.; Gordon, P.; Visich, P.; Keteyian, S. (2019). Clinical Exercise Physiology, 4th ed.; Human Kinetics: Champaign, IL, 2019; p. 383.

5. Faul, F.; Erdfelder, E.; Lang, A.G.; Buchner, A. G*Power 3: A flexible statistical power analysis program for the social, behavioral, and biomedical sciences. Behav Res Methods 2007, 39, 175-191.

6. Peacock, J.L.; Kerry, S.M.; Balise, R.R. Presenting medical statistics from proposal to publication. Oxford University Press, 2017; pp. 24-34.

7. Cohen, J.  A power primer. Psychol Bull 1992,112, 155-159.

8. Peacock, J.L.; Kerry, S.M.; Balise, R.R. Presenting medical statistics from proposal to publication. Oxford University Press, 2017; p. 50.

Reviewer 2 Report

Abstract is rather too long since a lot of statistic detail must appear in the "Results"

Figure 1 cation ought to be more informative.

More review of other related results could extend "Discussion"

Author Response

Responses to the reviewer 2  Comments

1. Reviewer: Abstract is rather too long since a lot of statistic detail must appear in the "Results".

Authors: We have deleted the statistical analysis sentence and removed some words (line 16).

2. Reviewer: Figure 1 caption ought to be more informative.

Authors: Revised and corrected.

3. Reviewer: More review of other related results could extend "Discussion".

Authors: Revised and corrected.

Round 2

Reviewer 1 Report

Thank you for your responses.